# Drug targeting CYP2E1 for the treatment of early-stage alcoholic steatohepatitis

Torsten Diesinger[1,2,13]*, Vyacheslav Buko[3,4], Alfred Lautwein[2], Radovan Dvorsky[5,6], Elena Belonovskaya[3], Oksana Lukivskaya[3], Elena Naruta[3], Siarhei Kirko[3], Viktor Andreev[7], Dominik Buckert[2,8], Sebastian Bergler[2], Christian Renz[2], Edith Schneider[9], Florian Kuchenbauer[9,10], Mukesh Kumar[11], Cagatay Günes[11], Berthold Büchele[12], Thomas Simmet[12], Dieter Müller-Enoch[2], Thomas Wirth[2], Thomas Haehner[2]

1 Donauklinik Neu-Ulm, Abteilung für Innere Medizin, Neu-Ulm, Germany, 2 Institute of Physiological Chemistry, Ulm University, Ulm, Germany, 3 Division of Biochemical Pharmacology, Institute of Biochemistry of Biologically Active Substances, Grodno, Belarus, 4 Department of Biotechnology, University of Medical Sciences, Bialystok, Poland, 5 Institut für Biochemie und Molekularbiologie II, Medizinische Fakultät der Heinrich-Heine-Universität Düsseldorf, Düsseldorf, Germany, 6 Max Planck Institute of Molecular Physiology, Dortmund, Germany, 7 Department of Medical Biology and Genetics, Grodno State Medical University, Grodno, Belarus, 8 Department of Internal Medicine II, University Hospital Ulm, Ulm, Germany, 9 Department of Internal Medicine III, University Hospital Ulm, Ulm, Germany, 10 University of British Columbia, Terry Fox Laboratory, Vancouver, Canada, 11 Department of Urology, University Hospital Ulm, Ulm, Germany, 12 Institute of Pharmacology of Natural Products and Clinical Pharmacology, University Ulm, Ulm, Germany, 13 Department of Internal Medicine, Neu-Ulm Hospital, Neu-Ulm, Germany

* tdiesinger@gmx.de

**Data Availability Statement:** All relevant data are within the manuscript and its Supporting Information files.

## Abstract

### Background and aims

Alcoholic steatohepatitis (ASH)—the inflammation of fatty liver—is caused by chronic alcohol consumption and represents one of the leading chronic liver diseases in Western Countries. ASH can lead to organ dysfunction or progress to hepatocellular carcinoma (HCC). Long-term alcohol abstinence reduces this probability and is the prerequisite for liver transplantation—the only effective therapy option at present. Elevated enzymatic activity of cytochrome P450 2E1 (CYP2E1) is known to be critically responsible for the development of ASH due to excessively high levels of reactive oxygen species (ROS) during metabolization of ethanol. Up to now, no rational drug discovery process was successfully initiated to target CYP2E1 for the treatment of ASH.

### Methods

In this study, we applied a rational drug design concept to develop drug candidates (NCE) including preclinical studies.

### Results

A new class of drug candidates was generated successfully. Two of the most promising small compounds named 12-Imidazolyl-1-dodecanol (abbr.: I-ol) and 1-Imidazolyldodecane (abbr.: I-an) were selected at the end of this process of drug discovery and developability. These new ω-imidazolyl-alkyl derivatives act as strong chimeric CYP2E1 inhibitors at a

**Funding:** The author(s) received no specific funding for this work.

**Competing interests:** The authors have declared that no competing interests exist.

nanomolar range. They restore redox balance, reduce inflammation process as well as the fat content in the liver and rescue the physiological liver architecture of rats consuming continuously a high amount of alcohol.

## Conclusions

Due to its oral application and therapeutic superiority over an off-label use of the hepatoprotector ursodeoxycholic acid (UDCA), this new class of inhibitors marks the first rational, pharmaceutical concept in long-term treatment of ASH.

## Introduction

Alcohol consumption is the world's third largest and Europe's second largest risk factor for disease burden [1]. The World Health Organization reports about two billion alcohol consumers worldwide and 76.3 million people with diagnosable alcohol use disorders. Alcohol abuse results in 3.3 million fatalities each year and contributes to 5.9% of all global fatalities [1, 2].

In the United States alone, alcoholic liver disease (ALD), which includes steatosis hepatis, alcoholic steatohepatitis (ASH) and liver cirrhosis, affects more than 2 million people (i. e. approximately 1% of the population). There is no FDA-approved therapy for ALD [3], although it accounts for the majority of chronic liver diseases in Western countries and remains a major cause of morbidity and mortality in the United States.

Excessive alcohol consumption causes enormous amounts of reactive oxygen species (ROS) in the liver, which cannot be balanced by the intracellular redox system anymore [4]. The consequence is a progression from steatosis hepatis to hepatocellular carcinoma (HCC).

Ethanol increases the enzymatic activity of cytochrome P450 2E1 (CYP2E1) via elevated gene expression and protein stabilization, which in turn increases the oxidation of ethanol to toxic acetaldehyde [5]. This pathway promotes an accelerated ethanol metabolism during chronic alcohol consumption and likely contributes to metabolic tolerance of alcohol addicts towards ethanol, which again enables further alcohol consumption. In addition, increased CYP2E1 activity releases ROS in a much higher level than all other hemoproteins of the gene family 2 and 4 [6]. This mechanism is thought to trigger ethanol-induced liver injury. Remarkably, CYP2E1 has normally a low enzymatic activity under physiological conditions, is not involved in any vital functions and is completely redundant [7]. There have been a few attempts in the last decades to develop CYP2E1 inhibitors aiming to reduce harmful effects of toxic metabolites caused by increased CYP2E1 activity. But so far, no drug candidate could be developed, which could successfully pass stages of discovery and developability, or even be applied in proof of concept studies concerning ASH.

It has been known for several decades that derivates of nitrogen containing heterocycles (pyridine, imidazole, triazole) are strong, reversible inhibitors of CYP450 isoenzymes [8]. Their specificity is due to the geometry and physicochemical behavior of the functional group of each heterocycle [9]. Among all 58 human CYP450 isoenzymes, only CYP4A11 and CYP2E1 are responsible for the ω-hydroxylation of medium-length chain carboxylic acids with dodecanoic acid (lauric acid) being metabolized most effectively [10]. Several independent studies could show that ω-imidazolyl-dodecanoic acid is the strongest inhibitor of CYP4A1 (ortholog for CYP4A11 in rattus norvegicus) among ω-imidazolyl-alkanoic acids and additional alkyl derivatives [11, 12]. Further studies of structure activity relationship (SAR) on CYP4A11 and its orthologs and CYP2E1 confirmed similarity in their potential to be inhibited

[10, 13, 14]. Thus ω-imidazolyl-alkyl derivatives seemed to represent a new class of drug candidates in targeting CYP2E1. Based on these considerations, we designed novel CYP2E1 inhibitors utilizing the concept of rational drug design. First we generated several hit compounds that were evaluated by in vitro inhibition experiments. In the second phase we compared the potency of these compounds to available, analogous compounds in subsequent in vitro studies. Most promising compounds were finally tested by in vivo studies in rats. At the end of the process, we selected two small compounds named 12-Imidazolyl-1-dodecanol (abbr.: I-ol) and 1-Imidazolyldodecane (abbr.: I-an).

Similar to other ω-imidazolyl-alkyl derivatives, both of these new inhibitors are assembled by two very different functional groups known to interact with CYP2E1. These are an imidazole ring acting as chelating agent and fatty acid derivatives that perform a molecular mimicry of natural substrates. The inhibitory potency of I-ol and I-an was very strong with a $K_i$ value in the nanomolar range. They restored intracellular redox balance via reduction of ROS and lipid peroxidation in cell culture as well as rats administered to daily alcohol consumption. Rats treated with either compound showed a lower serum and liver concentration of triglycerides, cholesterol and phospholipids compared to non-treated rats with ASH. The liver enzymes alanine aminotransferase (ALT), aspartate aminotransferase (AST), alkaline phosphatase (AP) and gamma-glutamyltransferase (γ-GT) had lost catalytic activities in the treatment groups. Histopathological sections confirmed the biochemical data showing an impressive similarity of liver architecture of healthy rats and those who were treated with I-ol and I-an. In comparison to the known hepatoprotector ursodeoxycholic acid (UDCA), which is the standard therapy for primary biliary cirrhosis (PBC) [15], primary sclerosing cholangitis (PSC) [16] and autoimmune cholangitis (AIC) [17], therapeutic effects of I-ol and I-an were stronger even at a 100-fold lower concentration.

These results point out a promising new opportunity in pharmacological therapy of ASH, which is attenuation of CYP2E1-caused oxidative stress.

## Material and methods

### Synthesis of ω-imidazolyl-alkyl derivatives (see US patent No. US 8153676 B2)

12-imidazolyl-1-dodecanol (I-ol) and 1-Imidazolyldodecane (I-an) were synthesized based on a method published by Lu et al. [18]. A mixture of 12-bromo-1-dodecanol and imidazole was heated to 80˚ C at a molar ratio of 1:3. for five hours. The raw product was extracted with dichloromethane/water. The organic phase was dried over $Na_2SO_4$ and concentrated by evaporation. 12-imidazolyl-1-dodecanol was recrystallized from benzene/n-hexane. The production of 7-Imidazolyl-1-heptanol, 9-Imidazolyl-1-nonanol and 10-Imidazolyl-1-decanol were performed at equimolar conditions with 7-bromoheptan-1-ol, 9-bromononan-1-ol or 10-bromodecan-1-ol respectively.

1-imidazolyldodecane was produced from 1-bromododecane and imidazole in a molar ratio of 1:3 while stirring and heating at 85˚ C. The raw product was dissolved in dichloromethane and poured out three times with water. The organic phase was dried over $Na_2S0_4$, filtered and concentrated by evaporation. The oily evaporation residue was induced to crystallize from n-hexane and yields 1-imidazolydodecane.

Phosphatidylcholine was reacted to form an O-phosphorylisourea under acidic conditions in the presence of dicyclohexylcarbodiimide. 12-imidazolyl-1-dodecanol was added to the reaction mixture. In so doing, dicyclohexylurea settled out. In order for this reaction to succeed, 4-diethylaminopyridine was necessary as the catalyst. The reaction mechanism is similar to that of the Steglich esterification, where dicyclohexylcarbodiimide is used in order to esterify an organic acid with an alcohol.

## Steady-state p-nitrophenol oxidation

Initial velocities of the oxidation of p-nitrophenol (p-NP) to p-nitrocatechol by human cytochrome P450 2E1 were determined by measuring the absorption at 535 nm or 546 nm to determine product formation as described before [19].

The reaction was performed in 100 mM potassium phosphate buffer pH 7.4, 3 mM $MgCl_2$ and 126 µM para-nitrophenol substrate. Alternatively, 100 mM HEPES pH 7.6 was additionally used for the enzyme kinetic measurements. A final concentration of either 50 nM enzyme in form of SUPERSOMES™ from BD Biosciences or microsomes in case of cell culture experiments was added. Reaction was initiated by adding 500 µM (enzyme kinetic measurements) or 0.1 mM (cell culture experiments) NADPH after preincubation at 37˚ C and stopped by adding 20% trichloroacetic acid (TCA) after 30 minutes to ensure linearity of product formation. The protein concentration for the microsomal fraction was determined by Bradford assay to calculate the results expressed as units of specific activity defined as the amount of enzyme that consumes 1 µmol/L of NADPH per hour per microgram of protein. Inhibitor stock solutions were prepared in 0.1% DMSO and diluted to a final DMSO concentration of 0.003% in the reaction mixture at most to avoid enzyme inhibition by this solvent. Inhibitory constants ($K_i$) were calculated by non-linear fitting of initial velocities. Maximum inhibition could not be achieved since final inhibitor concentrations were limited by DMSO.

## Difference spectra of inhibitors

Liver microsomes were prepared as follows: Rat liver was homogenized in 33 mM TRIS*HCl pH 7.4, 33 mM KCl, 1 mM EDTA, 250 mM sucrose and protease inhibitors. The homogenate was centrifuged first at 600 g then at 6,500 g for 10 minutes to separate nuclei and mitochondria. Supernatant was centrifuged at 105,000 g for 60 minutes. The pellet (microsomal fraction) was dissolved in 10 mM TRIS*HCl pH 7.4, 10 mM EDTA, 50% glycerol and stored at -80˚C [20].

Difference spectra measurements were performed in tandem cuvettes at 37˚C. The sample as well as the reference chamber contained rat liver microsomes in 100 mM HEPES buffer (pH 7.6). The baseline was recorded between 365 and 500 nm. Varying concentrations of 12-Imidazolyl-dodecan-1-yl-phosphocholine (5–95 µM) were added to rat microsomes. Difference spectra were obtained after the system reached equilibrium.

## Homology modeling, in silico ligand and protein preparation, docking

No structural information about the target protein CYP2E1 obtained by X-ray crystallography or NMR was available, when we started with our work on finding a new way for the treatment of ASH. To gain a deeper understanding of ligand-target interaction, we created ten molecular models of CYP2E1 (UniProt ID Q5VZD5[human]) derived from different X-ray structures of its closely related isoenzymes [21] that covered a broad conformational diversity—namely CYP2C5 (PDB IDs 1DT6[rabbit], 1N6B[rabbit], 1NR6[rabbit]), CYP2C9 (PDB IDs 1OG2 [human], 1OG5[human], 1R9O[human]), CYP2B4 (PDB IDs 1PO5[rabbit], 1SUO[rabbit], 2BDM[rabbit]) and CYP2D6 (PDB ID 2F9Q[human]). All CYP2E1 models were used for the subsequent molecular docking process. The ability of other imidazole-containing molecules to inhibit CYP2E1 was tested by searching for compounds with an imidazole ring in the ZINC database [17] (http://zinc.docking.org/). All commercially available compounds were docked into the CYP2E1 protein models. The overall inhibitory effect was evaluated as a sum of scoring values for best poses of each compound. The best ten compounds were purchased from the company Interbioscreen ltd (https://www.ibscreen.com/) for further in vitro enzymatic studies.

The software program Modeller [22] (https://salilab.org/modeller/) utilizing a method of homology modeling was used to create the structural models of CYP2E1. Preparations of protein and ligand as well as docking were further performed with software solutions from Schrödinger LLC (https://www.schrodinger.com/). First modeled structures were refined by 1,000 cycles of energy minimization using the module Macromodel and then processed using the Protein Preparation wizard. Ligands were constructed and prepared for docking with the help of the modules Build and LigPrep setting the pH value to $7.0 \pm 0.5$. Interactions of probe atoms with proteins for the purpose of docking were calculated with the Receptor Grid Generation command of Glide at the points of regular 3 D grid around the active site, spaced by 1 Å in all directions within a box of 40 Å site length and centered at the geometric center of heme molecules. To ensure that the ligands would bind near the Fe atom of heme, a distance constrain around 3.0 Å between the Fe atom and nitrogen atom of ligands was applied during the docking. Own docking was performed by command of Glide using the default parameter. Functional potency of molecules was assessed via scoring by Glide, which evaluates more promising molecules with lower values. Analysis of positions and conformations of docked molecules was done in PyMOL (https://pymol.org/2/).

During this study, newly solved crystal structures of human Cytochrome P450 2E1 have been determined in complexes with six different protein stabilizing ligands, whereas the complex forming moiety was always an imidazolyl group or a nitrogen containing aromatic ring [23, 24]. Three of the ligands were fatty acid-like derivatives that interacted with the iron ion of the 'open protein confirmation': imidazolyl-octanoic, -decanoic and -dodecanoic acid (PDB IDs 3GPH, 3KOH, 3LC4). The others were smaller chelating agents binding to a 'closed protein confirmation': 4-methylpyrazole, indazole and pilocarpine (PDB IDs 3E4E, 3E6I, 3T3Z).

We decided to verify our in silico calculations and performed an additional docking analysis with the ten best scoring library compounds and the six new protein structures following principally the same protocol as described above. Additionally, we were highly interested in the influence of the alkyl chain's length and degree of oxidation at the terminal carbon atom. So, we docked ω-imidazolyl-alkyl derivatives with alkyl chains varying from 5 to 13 carbon atoms combined with three 'head groups' (-H, -OH, -COOH) to each available crystal structure of 2E1.

## Chemical complexation of I-ol with HPβCD

12-imidazolyl-1-dodecanol (I-ol) was complexed with hydroxy-propyl-beta-cyclodextrin (HPβCD), in brief called cyclodextrin. To this end, I-ol and HPβCD were separately dissolved in ethanol. After mixing both solutions complex formation takes place under continuous stirring for 1 h at room temperature, followed by careful three step removal of the solvent (1st step: rotary evaporator; 2nd step: drying by nitrogen; 3rd step: lyophilization). The obtained complex, called I-ol-CD, with a molar ratio of 1:1.7 (I-ol: HPβCD) forms a clear solution by addition of water. Quality control as well as stability under various conditions was checked by HPLC and HPTLC analysis.

## Cell lines, culture conditions and microsomal preparation

HepG2 cells that do not express significant amounts of CYP2E1, CYP2E1 overexpressing HepG2 E47 cells as well as the non-CYP2E1-expressing HepG2 C34 control cell line transfected with an empty control plasmid were kindly provided by Dr. Arthur Cederbaum (Mount Sinai Health System, NY).

HepG2 cells were cultured in DMEM, supplemented with 10% fetal bovine serum (FBS), 100 U/mL penicillin, and 100 μg/mL streptomycin in a humidified atmosphere of 5% $CO_2$ at 37˚C.

OCI-AML3, MOLM-13, MOLP-8 cell lines were cultured in RPMI 1640 with 10% fetal bovine serum (FBS) and 1% penicillin-streptomycin. NB4 cells were cultured in RPMI 1640 with 20% fetal bovine serum (FBS) and 1% penicillin-streptomycin.

All cells were incubated with 40 μM I-ol complexed with CD (I-ol-CD) 72 h prior the experiment. I-ol-CD treated and untreated cells were washed twice with PBS and harvested by scraping. The pelleted cells were then resuspended in 1 ml ice cold Buffer A (33.3 mM $K_2HPO_4$, 33.3 mM KCL, 0.5 M EDTA, 1 M Saccharose, Protease Inhibitor) prior homogenization and disruption, which was performed in a glas douncer. Microsomes were prepared by differential ultracentrifugation, resuspended in Buffer A and kept at -80˚C.

## Animals, diets and treatment

The care and use of the animals and the procedures performed using them conformed to the institutional guidelines, national and international laws, and the guidelines set forth to Directive 2010/63/EU of the European Parliament and of the Council of 22.10.2010 on the protection of animals used for scientific purposes. The Local Ethics Committee for animal experiments in Grodno and the Ethic Committee of the National Academy of Sciences, Belarus approved the study.

I-ol and I-an were synthesized at the University of Ulm, Germany. Ursodeoxycholic acid (UDCA) was supplied by Prodotti Chimici a Alimentari S.p.a., Basaluzzo, Italy. Hydroxypropylmethylcellulose (hypromellose, HPMC) was purchased from Shin-Etsu Chemical Co. Ltd, Cellulose Division, Tokyo, Japan. All other chemicals used in this study were of finest analytical grade.

A total of 68 female Wistar rats (Institute of Pharmacology and Biochemistry Breeding House, Minsk, Belarus) aged 80–90 days and weighting 180–200 g at the beginning of the experiments were used in the study. Each rat group (n = 6–8) was housed in a separate cage and had free access to the liquid diet during the full study period. Prior to the study, the animals were habituated for seven days to the cage conditions and experimental handling. The rats were maintained in a controlled experimental environment (20–25˚C, 40–50% humidity, 12 h light per day, room with a 12-h light/dark cycle).

Rats were divided into ten groups characterized by the diet and the test compounds (S1 Table). Group 1 is the first control group in which a normal solid standard diet was fed for 12 weeks. All other groups received the liquid Lieber-DeCarli (LDC) diet [25] (S2 Table), which combines food and drink in a liquid manner. It was prepared by ssniff Spezialdiäten GmbH, Soest. The addition of maltodextrin to this diet should compensate the pure caloric effect of ethanol, whereby 1.77 g maltodextrin replaces 1 g ethanol. The calorie value of both the control and ethanol diets was the same: 35% calories from fat, 11% from carbohydrates, 18% from protein and 36% from ethanol.

In the first two weeks rats of all groups–with the exception of group 1 –were fed with LDC diet plus maltodextrin in an appropriate amount to adapt the animals to the liquid diet. Group 2 is the second control group in which the isocaloric diet was administered for further 10 weeks until the end of the experiment. The disease group corresponds to group 3 in which the amount of maltodextrin was gradually reduced in the first two weeks and replaced by a final amount of 5% ethanol (w/w). The test compounds I-ol, I-an and UDCA were dissolved in a 0.8% aqueous HPMC gel to obtain the appropriate concentrations: (1) I-ol with 0.4, 4 and 40 mg/kg b. w. (2) I-an with 0.4, 4 and 40 mg/kg b. w. (3) UDCA with 40 mg/kg b. w. Rats of group 2 and 3 received only HPMC as a placebo. The LDC diets were fed ad libitum and their intake per cage was controlled daily. The corresponding bottles were stored in the cages from 08:00 a. m. until 06:00 p. m. and replaced by drinking water bottles overnight until 08:00 a. m.

in the next morning. Every day at the same time, the test compounds or only HPMC were administered intragastrically by oral gavage with an administration volume of 2 ml/kg b. w. After ten weeks of treatment, the rats were anesthetized with an intraperitoneal injection of a 5% pentobarbital solution and then killed by aorta scarification. Their livers were dissected and liver pieces frozen and stored in liquid nitrogen. A part of the liver was fixed for histological investigations. Blood was collected before sacrifice, centrifuged for serum preparation and finally stored in liquid nitrogen.

## Measurement of oxidative stress in liver and cell culture

The generation of reactive oxygen species (ROS) in rat liver was evaluated by NADPH-induced chemiluminescence with amplification by both luminol for the detection of superoxide (SRA) radical anion and lucigenin detecting hydrogen peroxide as described by Müller-Peddinhaus [26]. In short, liver pieces were homogenized in 0.1 M sodium phosphate buffer pH 7.4. The homogenates were centrifuged at 9,000 x g at 0˚C and aliquots were used for the measurement. Reduced glutathione (GSH) was determined by a modified Ellman's method [27]: 0.2 ml of the liver homogenates' supernatant were added to 1.6 ml of an EDTA-/sodium phosphobenzoic acid solution. The absorbance was read at 412 mm against blanks. GSH solution—prepared immediately before use—served as a standard. End products of the lipid peroxidation reaction are equivalent to TBARS (thiobarbituric acid-reacting substances) and their concentration was measured according to Buege et al. [28]. The enzymatic activity of cytosolic glutathione peroxidase (GPx) and microsomal glutathione reductase (GR) were determined as described by Buko at al. [29]. Catalase activity was evaluated by spectrophotometric measurement using hydrogen peroxide as a substrate according to Beers et al. [23]. 100,000 cells were seeded in 24-well plates and incubated for 72 hours with 40 μM I-ol complexed with CD (I-ol-CD) or as negative controls without any supplement. 40 μM I-ol-CD were administered to three different HepG2 cell types: (1) native HepG2 cells, (2) the E47 cell line constitutively and stably overexpressing CYP2E1 protein (3) the C34 cell line stably maintaining an anti-sense CYP2E1 cDNA, and hence unable to produce the protein product. Three different AML cell lines (OCI-AML3, NB4, MOLM-13) and one myeloma cell type (MOLP-8) not known for expression of CYP2E1 and without any importance in CYP2E1 mediated ROS production were used as control group. For the detection of ROS, cells were trypsinized, washed and incubated in 1 ml culture media supplemented with 10 μl of 5 mM Dichlorofluorescein-diacetat (DCFH-DA) in the dark for 1 hour at 37˚C. After incubation, the cells were washed twice in PBS and resuspended in SYTOX before immediate analysis on a Fortessa flow cytometer at 488 nm excitation and 525 nm emission wave length.

## Serum specific parameters

The serum marker enzyme activities, alanine aminotransferase (ALT), aspartate aminotransferase (AST), alkaline phosphatase (AP) and gamma-glutamyltransferase (γ-GT) as well as triglycerides and bilirubin were measured according to manufacturer's instructions using standard commercial kits from Lachema (PLIVA-Lachema Diagnostika, Czech Republic). Serum concentrations of leptin, adiponectin and TNF-alpha were analyzed using commercial ELISA test kits: BioCat GmbH (Heidelberg, German) for adiponectin and TNF-alpha, BioVendor Laboratory Medicine Inc. (Brno, Czech Republic) for leptin. Insulin serum concentration was determined by radioimmunoassay using the commercially available kit RIO-insulin-PG-J125 (IBOCH, Minsk, Belarus). Glucose serum concentration was determined by the Accu-Check Active blood glucose testing system from Roche-Diagnostics GmbH (Mannheim, Germany).

## Lipid-related parameters

Lipids were extracted with a chloroform: methanol mixture (2:1 v/v). Neutral lipids and phospholipids were separated into classes by thin-layer chromatography (TLC). Lipid fractions (triglycerides, cholesterol and its fractions) were measured in extracts from scraped spots by routine methods using the appropriate commercial kits. Triglycerides and cholesterol measurements, respectively those from Lachema (Brno, Czech Republic), were used according to manufacturer's instructions. Phospholipids were measured according to a method based on the determination of inorganic phosphate with molybdate reagent [24]. VLDL were detected by the manganese precipitation method [30], which was modified [31] in part. Cholesterol from LDL and HDL was measured in lipid extracts by routine methods according to manufacturer instructions using commercial kits from Lachema (Brno, Czech Republic).

## Fibrosis parameter of the liver

Total hydroxyproline contents were analyzed after digestion with acid [32]. Pieces of the liver were used for a Mallory-Azan (Heidenhain's Azan) staining in connection with histological preparations of the liver samples. This connective tissue stain is a modification of Mallory's original connective tissue stain, in which azocarmine is used along with the aniline blue-orange G mixture. Collagen and basophil granules stain 502 blue, muscle and acidophil granules stain orange to red and nuclei and cytoplasm stain red. Elastic fibers are unstained or stain yellow or pink. The fraction of positively stained liver tissue related to the histological slides' area was determined semiquantitatively.

## Liver histopathology

Liver samples were selected randomly, fixed in Bouin's solution and embedded in paraffin wax. Histological sections were prepared and stained with haematoxylin and eosin. Other liver samples were fixed in a solution containing formaldehyde, paraformaldehyde and glutaraldehyde with a subsequent post-fixation in 1% osmium containing phosphate buffer pH 7.4 for further analysis. Liver tissue was embedded in a mixture of butyl-methyl-metacrylates. Semifine sections of 0.5–1.0 μm were stained with Asur II, methylen blue and basic fuxine.

## Statistical analysis

We used SPSS as software tool for statistical analysis and GraphPad PRISM to create the diagrams.

Concerning cell culture experiments, results are shown as means ± SDs. Statistical evaluation was carried out by two-way ANOVA followed by a Bonferroni post-hoc test. Results of the animal study are depicted as individual values of the dependent variable (i. e. the measured pathological parameters) grouping around the mean value depicted as horizontal line. Statistical calculation was performed based on one-way ANOVA and subsequent complex custom/planned contrasts, thereby grouping the contrasts as follows: statistical difference between (a) diet groups (group 1–3), (b) I-ol groups (group 4–6) and disease group (group 3), (c) I-an groups (group 7–9) and disease group (group 3), (d) UDCA (group 10) and disease group (group 3). A simple contrast analysis was performed only if no statistical average difference between the therapy groups (I-ol and I-an) and the disease group was measured. This allowed us to evaluate a statistically relevant influence of each inhibitor concentration when the whole group failed.

Due to a clear assumption about the effect of the different animal diets (group 1–3) and the tested compounds (group 4–10), hypotheses were considered as unilateral. In all cases the very

conservative correction of the type I error regarding multiple comparisons was performed by Bonferroni adjustment lowering the significance level to * $p < 1.67E-02$ (significant); ** $p < 3.33E-03$ (very significant); *** $p < 3.33E-04$ (extremely significant).

Dose-dependent effects of I-ol as well as I-an on each clinical parameter were calculated by Pearson's correlation and linear regression analysis with the following statistical parameters: + $p < 0.05$ (significant); ++ $p < 0.01$ (very significant); +++ $p < 0.001$ (extremely significant); $0.10 \leq |r| < 0.30$ (moderate correlation); $0.30 \leq |r| < 0.50$ (moderate correlation); $|r| \geq 0.50$ (strong correlation); $0.0196 \leq R^2 < 0.1300$ (small effective power); $0.1300 \leq R^2 < 0.2600$ (moderate effective power); $R^2 \geq 0.2600$ (strong effective power).

## Results

### In silico and in vitro target-inhibitor interactions

Before starting the time-consuming and expensive drug discovery process, information about the transferability of results from animal species to human must be gained. For this purpose, we compared the CYP2E1 amino acid sequence of several species (S1 Fig). In comparison to the complete human CYP2E1 sequence, animal sequences showed an average identity of 79%. Considering only residues interacting with the ligands/inhibitors inside the active site, such identity increased to 84% (distance of 5 Å) or 86% (distance of 4 Å) (S3 Table). These numbers indicate that amino acids of CYP2E1, which are important for its enzymatic function, are highly conserved among mammals. So observations obtained for one species are likely to be also relevant for another species.

Knowing that lauric acid is most effectively metabolized by CYP2E1 and ω-imidazolyl-dodecanoic acid is the strongest inhibitor of CYP4A11 and its orthologs, we started the ligand-based drug design process with the ω-imidazolyl-dodecane derivatives I-ol, I-an and I-phosphocholine. These have an alkyl chain with twelve carbon atoms and an imidazole ring, which differs only in the chemical substitution at the carbon atom C1 (in the case of I-an C12). With a $K_i$ value of 121 nM to 612 nM, I-ol showed the strongest inhibitory effect of all small experimental substances studied in this scientific paper. The exchange of the polar head group with a hydrogen atom led to a lower affinity of I-an. The substitution of a bulky phosphocholine group showed the lowest inhibition effect with a $K_i$ of 22 μM. In accordance with the fitted equations, all three compounds showed an equivalent inhibitory mechanism using SUPERSO-MES[TM] as a model system. I-ol thus acted as the strongest complexing agent within the coordination sphere of porphyrin (Fig 1A and 1B).

Next, we reduced the alkyl chain length from I-ol to 10, 9 and 7 carbon atoms (Fig 1C). The inhibitory effect of 10-imidazolyl-1-decanol (abbr. I-decanol) and 9-imidazolyl-1-nonanol (abbr. I-nonanol) was almost identical, but 7-imidazolyl-1-heptanol (abbr. I-heptanol) showed a dramatic decrease in inhibitory performance. It was not possible to calculate a reliable kinetic parameter, since the necessary concentrations of the solvent DMSO have an inhibitory effect on CYP2E1 (doi: 10.6084/m9.figshare.12387107, https://figshare.com/s/f3ec912565aee777ea06). This finding is in line with the observation of Porubsky et al. [33], in which a missing spin shift by ω-imidazolylhexanoic acid will be reported.

To determine whether imidazole-containing compounds are already commercially available and would be a promising candidate for drug development, we conducted a large virtual library search for synthetic compounds (http://www.d4sky.eu/data/sc_out_with_scores.xlsx, or S1 File) and natural compounds (http://www.d4sky.eu/data/nc_out_with_scores.xlsx, or S2 File). The top ten compounds were tested in vitro (S5 Table), but none of these library compounds showed a greater inhibitory potential than I-ol (S2 Fig).

I-phosphocholine was used for all synthesized ω-imidazolyl-alkyl derivatives to illustrate the binding mode in the active center of CYP2E1 (Fig 1D). The typical type II difference

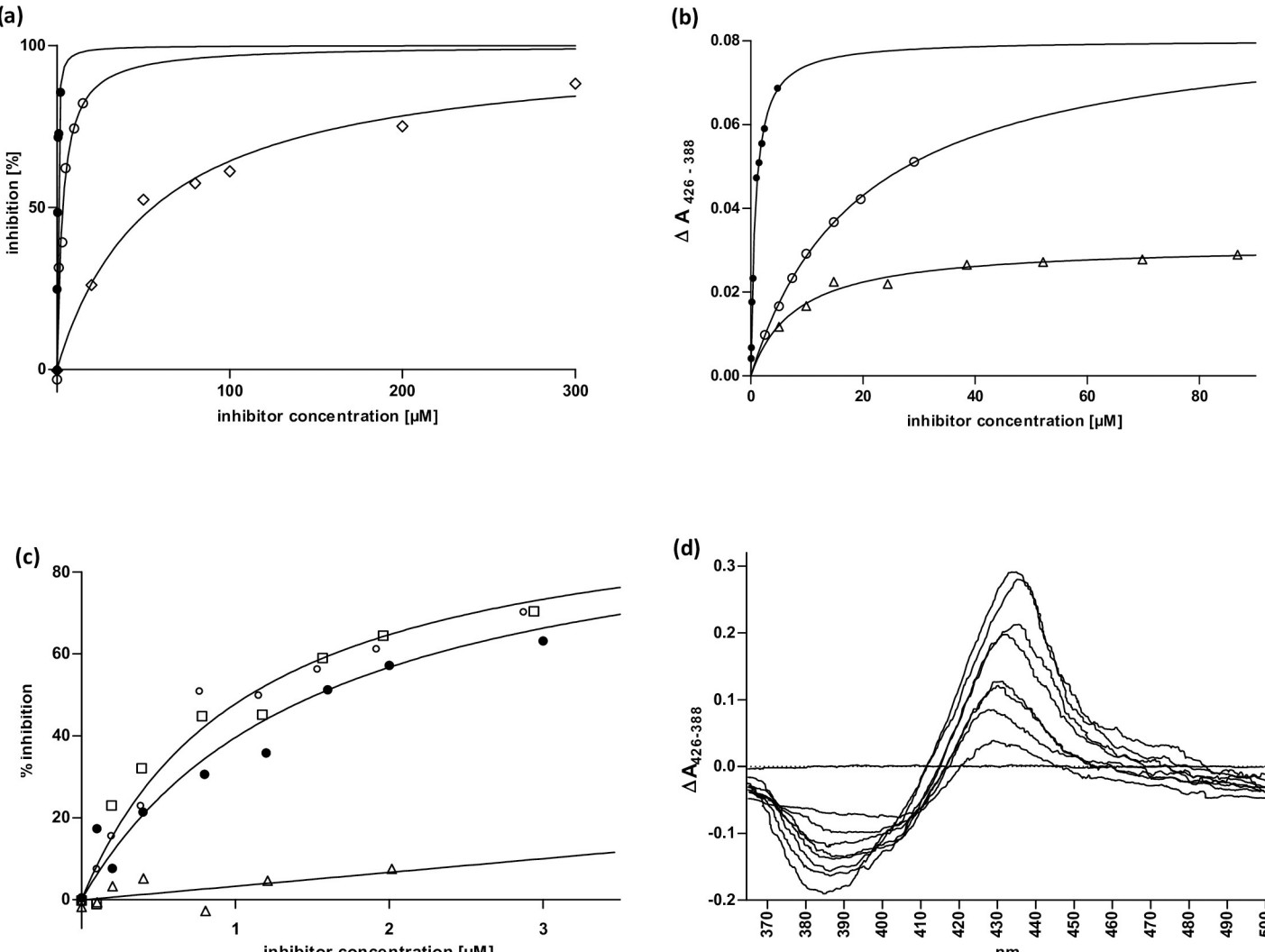

**Fig 1. Enzyme kinetics and ligand binding in subcellular systems (microsomes + supersomes). (a)** CYP2E1 inhibition by 12-imidazolyl-dodecane derivatives. Varying concentrations of the inhibitors 12-Imidazolyl-dodecanol (I-ol) (•), 1-Imidazolyl-dodecane (I-an) (○) and 12-Imidazolyl-dodecan-1-yl-phosphocholine (I-phosphocholine) (◇) were added to human CYP2E1 reconstituted system to calculate $K_i$ values. Nonlinear fitting was achieved using competitive inhibition as model including an offset in the case of I-phosphocholine **(b)** $K_D$ of imidazolyl-dodecane derivatives. Difference spectra of rat liver microsomes titrated with the inhibitors I-ol (•), I-an (○) and I-phosphocholine (△) were recorded limited by the solubility of the reagents. Experimental data were fitted to an equation including one specific binding site. **(c)** CYP2E1 inhibition by ω-imidazolyl-alkyl derivatives. Varying concentrations of the inhibitors 7-Imidazolyl-heptanol (△), 9-Imidazolyl-nonanol (□), 10-Imidazolyl-decanol (○) and I-ol (•) were added to a human CYP2E1 reconstituted system in supersomes to calculate $K_i$ values. Nonlinear fitting was achieved using competitive inhibition as model. **(d)** Difference spectra of 12-Imidazolyl-dodecan-1-yl-phosphocholine. It showed a characteristic Soret peak at 425–435 nm and minimum at 390–410 nm, which corresponds to a typical type II difference spectrum and is induced by substances both providing a bond to the trivalent heme iron (III) and interacting with the active center of the enzyme.

spectrum [34] led to a shift from high spin to low spin with an interruption of the catalytic cycle, corresponding to a loss of function. Both mechanisms–the steric blockade of and the lack of electron transfer to the CYP450 isoenzyme–turn substances such as ω-imidazolyl-alkyl derivatives into very strong reversible inhibitors.

We re-evaluated the in silico calculations with the newly resolved crystal structures of human CYP2E1 (S3 Fig). The general superiority of alkyl derivatives was confirmed. This could result from its molecular flexibility, which leads to a maximization of the binding energy through hydrophobic interactions. In addition, the head group could lead to an optimization

of the molecular interactions with corresponding side chains of the protein. These effects increased with increasing length of the alkyl chain and explained the in vitro observation that the inhibition of the catalytic activity of CYP2E1 measured with I-heptanol was significantly lower than with I-nonanol, I-decanol or I-(dodecane)ol.

## I-ol suppresses CYP2E1 activity and ROS production of a stably overexpressing HepG2-cell line

Computer simulation experiments showed that hydroxy-propyl-beta-cyclodextrin (HPβCD) seemed to be an excellent carrier for I-ol (S4 Fig). The water-soluble complex of HPβCD and I-ol (abbr. I-ol-CD) was produced in a molar ratio of 1:1.7, indicating an exceptionally strong interaction between the two molecules in aqueous solution. The results achieved with HPβCD as a carrier substance for I-ol fulfilled all requirements for subsequent in vitro and in vivo investigations.

Since I-ol bound to HPβCD showed a very strong inhibitory effect on CYP2E1 activity and advantageous binding properties, we only used this lead compound for further in vitro analyses, i.e. to inhibit cellular CYP2E1 activity and subsequent ROS formation.

As expected, the CYP2E1 overexpressing cell line E47 showed a significant increase in enzymatic activity and a significantly increased ROS production. Native HepG2 and C34 cells showed only weak enzymatic activity and an exceptionally low ROS production. The activity in the cell types AML and myeloma was below the detection limit of the assay, while ROS production could be detected (Fig 2C), probably by other means. I-ol-CD was able to reduce the amount of enzymatic activity of CYP2E1, which is significantly overexpressed in E47 cells, as well as its ROS production overexpressing E47 cells in a very significant way (Fig 2A and 2B).

For all other cell types, I-ol-CD had no influence on both parameters. The lack of inhibitory effect of I-ol-CD in HepG2 and C34 cells is due to enzymatic activity of most CYP450 isoenzymes, especially CYP2E1, in cultured liver cells. This situation relates in particular to liver cancer cell lines [35]. This means that the difference in enzymatic turnover of p-nitrophenol caused by CYP2E1 was below the detection limit. On the other hand, the substrate p-nitrophenol used as a probe for the enzymatic activity of CYP2E1 is not exclusively specific for CYP2E1. It can also be metabolized by other isoenzymes (e. g. CYP2A6, CYP2C19, CYP3A) [36, 37].

In summary, I-ol-CD strongly and significantly prevented ROS production by in vitro inhibition of the enzymatic activity of CYP2E1 in cells with elevated CYP2E1 levels.

## I-ol and I-an prevent ROS generation in the liver of rats

The pathogenesis of alcoholic steatohepatitis is related to the formation of a harmful amount of ROS. Therefore, the reduction of increased ROS production by inhibition of CYP2E1 catalysis is the starting point in the therapy of ASH.

Although so far only I-ol has been tested at the cellular level, we have decided to use I-an additionally as a backup compound for the planned proof of concept study in rats.

I-ol and I-an reduced the ethanol effect on the amount of **superoxide radical anion (SRA)** (Fig 3A) by an average of 80% and 96%, respectively, corresponding to a more than complete elimination of the ethanol effect. UDCA showed no statistic significant reduction.

I-ol and I-an not only completely reduced the ethanol effect on the amount of **hydrogen peroxide** (Fig 3B), I-an reduced the amount by 56% below the reference value. UDCA showed a 46% elimination of the ethanol effect.

I-ol and I-an reduced the ethanol effect on the amount of **reduced glutathione (GSH)** (Fig 3C) by an average of 36% and 55%, the highest concentration of I-an increasing the GSH value

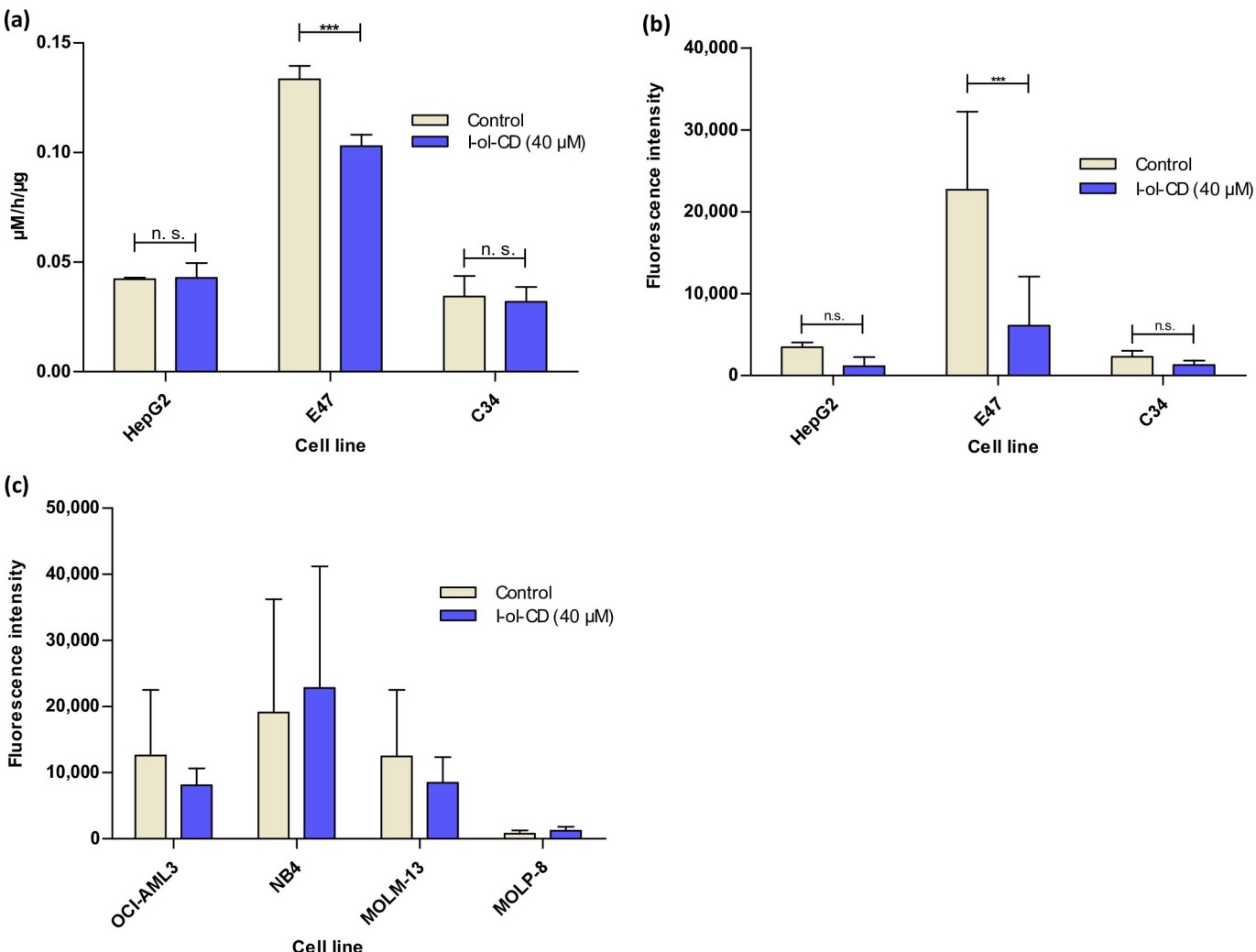

**Fig 2. Cell culture effects of I-ol.** Three hepatocyte cell lines (HepG2 cells, E47 cells and C34 cells) as well as three AML cell lines (OCI-AML3, NB4 and MOLM-13) and one myeloma cell line (MOLP-8) were incubated with 40 μM I-ol complexed with HPβCD (I-ol-CD) 72 h prior to the experiment. n = 3 independent experiments **(a)** Specific enzyme activity of CYP2E1 in hepatocytes. **(b)** ROS production in Hepatocytes. **(c)** ROS production in AML and myeloma cell lines. These cell lines served as negative control. Results are shown as means ± SDs. Statistical evaluation was carried out by two-way ANOVA followed by a Bonferroni post-hoc test. * p < 0.05 (significant); ** p < 0.01 (very significant); *** p < 0.001 (extremely significant).

by 2% above the reference value measured in healthy animals. UDCA displayed no statistically significant changes.

I-an reduced the ethanol effect on the amount of **thiobarbituric acid reactive substances (TBARS)** (Fig 3D) by an average of 93%, the highest concentration of I-an and I-ol lowering the TBARS level by 31% and 11% respectively below the reference values measured in healthy animals. UDCA displayed no statistic significant reduction.

On average, I-ol and I-an completely reduced the effect of ethanol on the enzymatic activity of **glutathione peroxidase (GPx)** (Fig 3E), in particular their highest concentration reducing it by 8% and 10% below the reference value in healthy animals. UDCA even displayed a stronger reduction of the ethanol effect by 28% below the reference value.

I-ol and I-an reduced the ethanol effect on the enzymatic activity of **glutathione reductase (GR)** (Fig 3F) by an average of 41% and 43%, respectively. UDCA showed no statistically significant result.

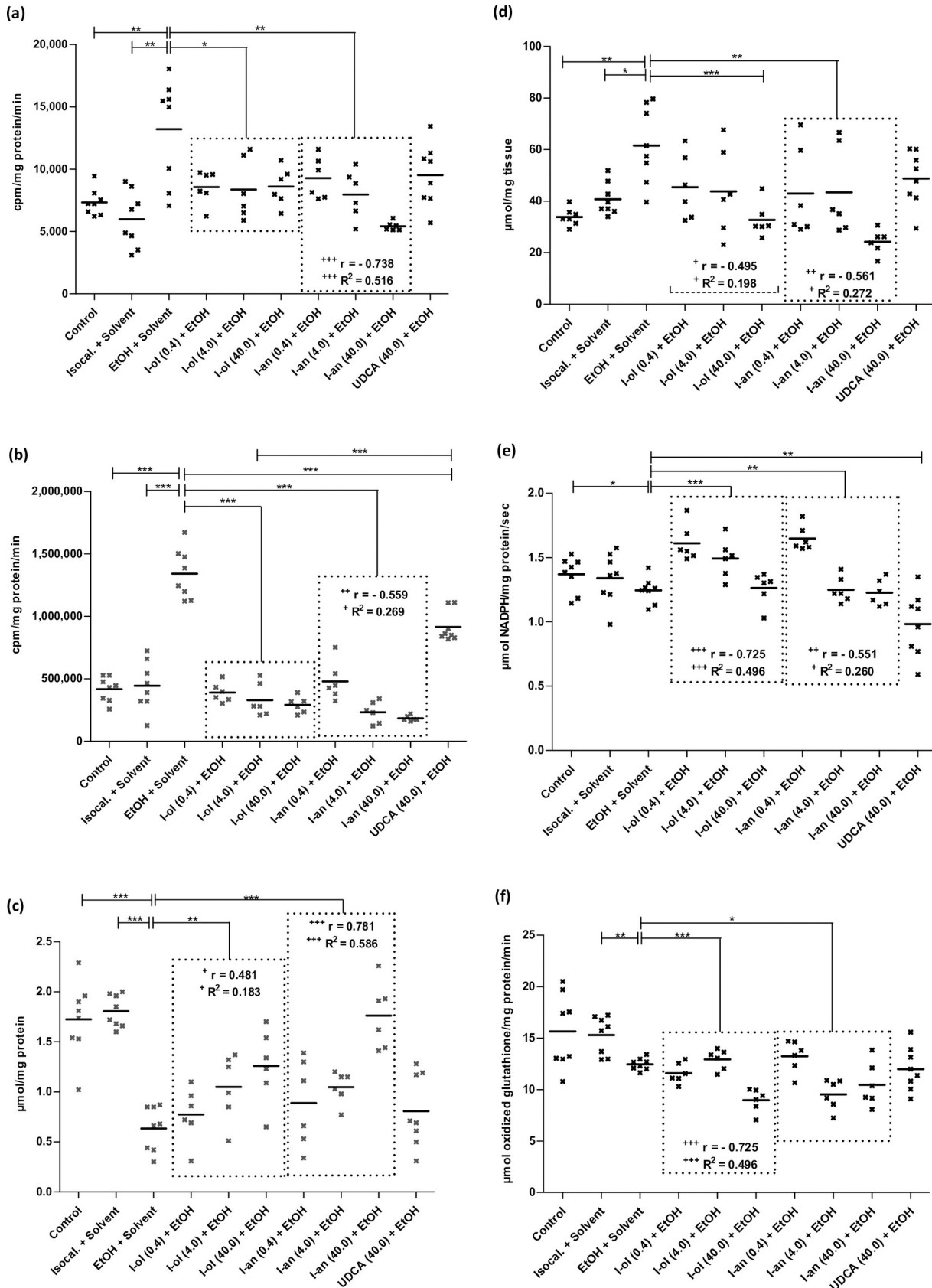

**Fig 3. In vivo ROS state in rat liver. (a)** SRA is elevated by 80% and 121% respectively in ethanol-fed rats compared to control groups 1 and 2. Compared to the disease group, I-ol and I-an cause an average decrease by 36% and 43% respectively. **(b)** Hydrogen peroxide is elevated by 221% and 202% in ethanol-fed rats compared to control groups 1 and 2. Compared to the disease group, I-ol and I-an cause an average decrease by 75% and 78%. UDCA displays a decrease of 32%. **(c)** GSH is decreased by 63% and 65% in ethanol-fed rats compared to control groups 1 and 2. Compared to the disease group, I-ol and I-an elevate the average amount of GSH by 62% and 94% respectively. **(d)** TBARS is increased by 76% and 51% in ethanol-fed rats compared to control groups 1 and 2. Compared to the disease group, I-an shows an average decrease of 40% and I-ol at its highest concentration by 50%. **(e)** GPx activity is decreased by 9% in ethanol-fed rats compared to control group 1. Compared to the disease group, I-ol and I-an elevate the average activity by 17% and 10%, respectively. UDCA shows a 21% decrease in activity. **(f)** GR activity has decreased by 19% in ethanol-fed rats compared to control groups 2. Compared to the disease group, I-ol and I-an reduce enzymatic activity by 10% and 11% and increase ethanol effect by 41% and 43% respectively. Rats from the disease group (n = 8) as well as from the treatment groups (n = 6 to 8) could not avoid daily alcohol intake because 50 g/l alcohol was mixed in water. Each point represents the individual value of the pathologically relevant parameter of a single rat. The mean value is displayed as horizontal line. Statistical calculation was based on one-way ANOVA and subsequent complex planned contrast or simple contrast analysis. Bonferroni adjustment for multiple comparisons reduced the significance level to $^*$ $p < 0.0167$; $^{**}$ $p < 0.00333$; $^{***}$ $p < 0.000333$. Dose-dependent effects were specified by Pearson's correlation coefficient r and linear regression analysis by the coefficient of determination $R^2$ with $^+$ $p < 0.05$; $^{++}$ $p < 0.01$; $^{+++}$ $p < 0.001$.

A significant reduction of the enzymatic activity of glutathione peroxidase (GPx) and gluta-thione reductase (GR) in chronic alcohol consuming animals was also observed in rats [38] and in serum of human patients [39] by other authors. The administration of I-ol and I-an in their lowest concentrations led to a sharp increase in the enzymatic activity of GPx, which could be correlated with a restoration of transcriptional regulation. Higher doses lowered the activity to a level below that of the control groups, leading to a complete lack of activation of ROS. Comparable results were observed for GR, even at the lowest concentrations of I-ol and I-an. UDCA had its strongest effect on GPx but none on GR.

In summary, I-ol and I-an almost prevented the generation of ROS, eliminated the ethanol effect and restored the redox balance in the liver in an extremely significant way, while UDCA had only a minor effect on hydrogen peroxide levels.

## I-ol and I-an restore the lipid balance

Degenerative changes of the fatty liver is an absolute prerequisite for the development of ASH.

I-ol and I-an reduced the ethanol effect on the concentration of **hepatic triglycerides** (Fig 4A) by an average of 71% and 73%, respectively. UDCA diminished the ethanol effect by 55%.

I-ol reduced the ethanol effect on the concentration of **hepatic phospholipids** (Fig 4B) by an average of 65%. I-an and UDCA completely eliminated the ethanol effect.

I-an reduced the ethanol effect on the concentration of **hepatic cholesterol** (Fig 4C) by 82%, with UDCA showing an almost identical result with a 78% elimination of the ethanol effect. I-ol had no influence on this parameter.

I-ol and I-an reduced the ethanol effect on the concentration of **serum triglycerides** (Fig 4D) by an average of 91% and 83% respectively, which corresponded to an almost complete elimination of the ethanol effect. I-an showed a significant linear dose-dependent effect. UDCA showed a 79% reduction in ethanol effect.

I-ol and I-an reduced the ethanol effect on the **serum VLDL** concentration (Fig 4E) by 55% and 59% and thus halved it. UDCA showed no statistically significant reduction.

I-an reduced the ethanol effect on the concentration of **serum LDL cholesterol** (Fig 4F) by 70%. I-ol at its lowest concentration decreased it by 61%. UDCA displayed no significant effect.

In summary, the therapeutic effects of I-ol and I-an on lipid metabolism were very pro-nounced. Both drug candidates and UDCA led to a strong reduction of the pathophysiologi-cally elevated lipid parameters in liver and serum. I-ol an I-an showed an almost complete restoration of normal triglyceride levels in the liver and generally a higher statistical

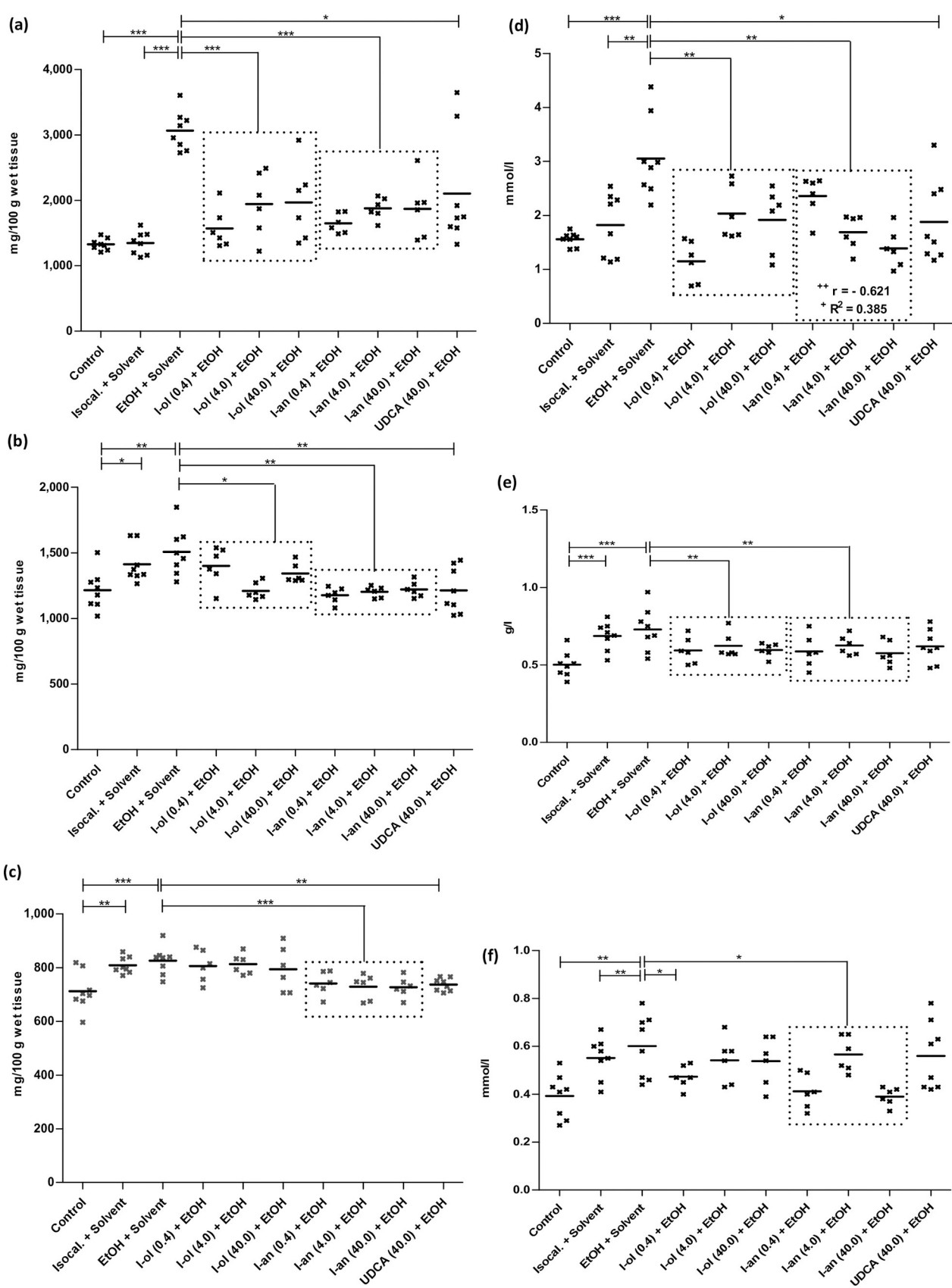

**Fig 4. Liver and serum fat content. (a)** The triglycerides of the liver are elevated by 131% and 127% compared to control groups 1 and 2. Compared to the disease group, I-ol and I-an cause an average decrease by 40% and 41% respectively. UDCA displays a decrease by 31%. **(b)** Liver phospholipids are elevated by 24% compared to control group 1 but no statistically significant changes are observed compared to control group 2. Compared to the disease group, I-ol and I-an cause an average decrease of 13% and 20% respectively. UDCA displays a decrease by 20%. **(c)** The liver cholesterol level is increased by 16% compared to control group 1. Compared to the disease group, I-an and UDCA cause a decrease of 11%. I-ol has no influence on this parameter. **(d)** The serum triglycerides are elevated by 96% and 68% compared to control groups 1 and 2. Compared to the disease group, I-ol and I-an show an average decrease of 44% and 41% respectively. UDCA has a 39% reduction. **(e)** Serum VLDL is increased by 45% compared to control group 1. In comparison to the disease group, I-ol and I-an show an average decrease by 17% and 18% respectively. **(f)** The LDL cholesterol level in serum is elevated by 53% and 9% compared to control groups 1 and 2. Compared to the disease group, I-an shows an average decrease of 24%. Animals were treated in groups and statistical data evaluation was done as described in Fig 3.

significance compared to UDCA. I-an showed a strong linear dose-dependent effect on triglyceride serum concentration.

## I-ol and I-an restore the activity of liver enzymes

I-ol and I-an reduced the ethanol effect on the enzymatic activity of **ALT** (Fig 5A) by an average of 70% and 59% respectively. I-ol in the lowest concentration led to a complete restoration of the basic activity of healthy animals.

I-ol and I-an reduced the ethanol effect on the enzymatic activity of **AST** (Fig 5B) by an average of 60% and 48%, respectively, and thus approximately halved the ethanol effect. The lowest concentration of both compounds decreased AST activity to only 6% above the reference value measured in healthy animals. UDCA displayed an 83% reduction which correlated with an almost complete elimination of the ethanol effect.

In this study, the activity of ALT and AST increased by 30% and 71% %, respectively, over control group 1, while the De-Ritis quotient was 1.6 indicating severe liver damage in the disease group. All three compounds were able to reduce these enzymatic activities significantly or even very significantly. Surprisingly, it turned out that the therapeutic effect decreased with increasing concentration of I-ol and I-an. This trend was statistically significant for I-ol.

I-ol and I-an reduced the ethanol effect on the enzymatic activity of **γ-glutamyltransferase (γ-GT)** (Fig 5C) by 68% and 60%, respectively, which more than halved the ethanol effect. The highest concentration of both compounds decreased γ-GT activity, which was 28% and 12% above the reference value measured in healthy animals. UDCA displayed a 61% reduction of the ethanol effect.

I-an on average and I-ol at its lowest concentration reduced the ethanol effect on the enzymatic activity of **alkaline phosphatase (AP)** (Fig 5D) by 96% and 87% respectively, corresponding to an almost complete elimination of the ethanol effect. UDCA displayed a decrease by 66%.

In summary, the generally very significant increase of all four liver enzyme activities could be almost completely reduced by all tested compounds in a generally very or extremely significant way. There was a trend towards a maximum effect at the lowest concentration of I-ol and I-an.

## I-an and I-ol prevent ethanol-induced liver injury (histopathology)

Fat deposits often show a picture of macrovesicular steatosis with large cytoplasmic lipid vacuoles displacing the nucleus to the periphery. However, the morphological picture of steatohepatitis does not always correlate with the measured clinical parameters. It can contain both minimal damages and necroinflammatory processes.

Ethanol caused significant histopathological changes in the disease group (Fig 6C). Intracytoplasmic lipid vacuoles appeared very impressively in the form of macrovesicular steatosis with focal necrosis and intralobular leucocyte infiltration. An increase in connective tissue was

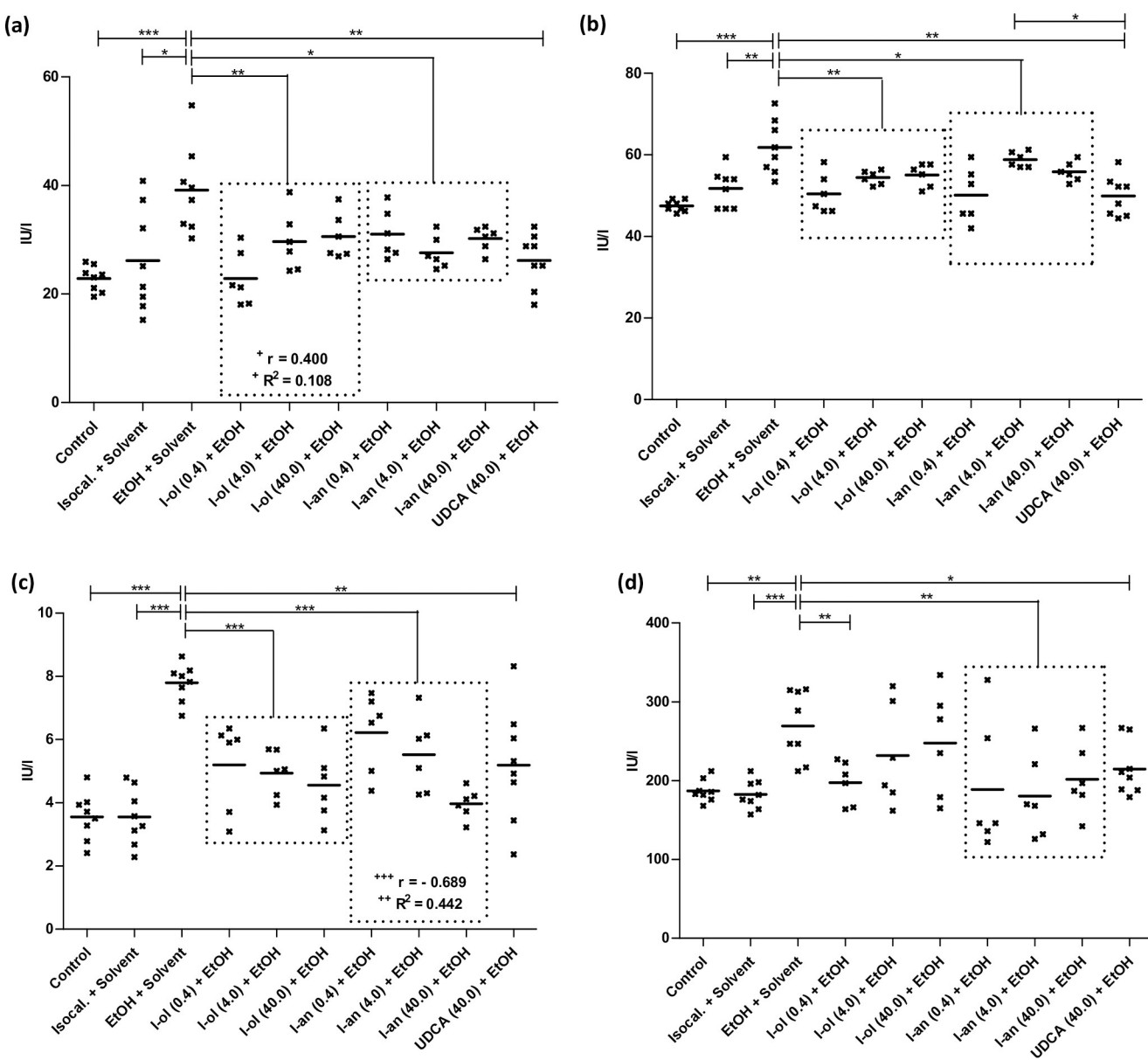

**Fig 5. Enzymatic activity of liver enzymes. (a)** ALT is elevated by 71% and 50% compared to control groups 1 and 2. Compared to the disease group, I-ol and I-an show an average decrease in enzymatic activity by 29% and 24%, respectively. UDCA shows a decrease by 33%. **(b)** AST is elevated by 30% and 19% respectively compared to control groups 1 and 2. Compared to the disease group, I-ol and I-an show an average decrease in enzymatic activity of 14% and 11% respectively. UDCA displays a decrease of 19%. **(c)** γ-GT is elevated by 119% over control groups 1 and 2, with I-ol and I-an showing an average decrease in enzymatic activity of 37% and 33% respectively over the disease group. UDCA displays a decrease of 33%. **(d)** AP is elevated by 44% and 48% respectively compared to control groups 1 and 2. Compared to the disease group, I-an shows an average decrease in enzymatic activity of 29%. I-ol is able to reduce the activity in its lowest concentration by 27%. UDCA shows a decrease by 20%. Animals were treated in groups and statistical data evaluation was done as described in Fig 3.

observed (S7 Table), but this was not significant. For this reason, the animal model used reflects an early stage of ASH, where the fibrotic remodelling process is still in its early development. The administration of I-ol and I-an in their higher concentrations led to an almost complete restoration of the physiological liver morphology (Fig 6D–6I). UDCA was able to reduce the histopathological effects of ethanol in a comparable way (Fig 6K). The result of the semi-quantitative analysis is summarized in S6 Table [40].

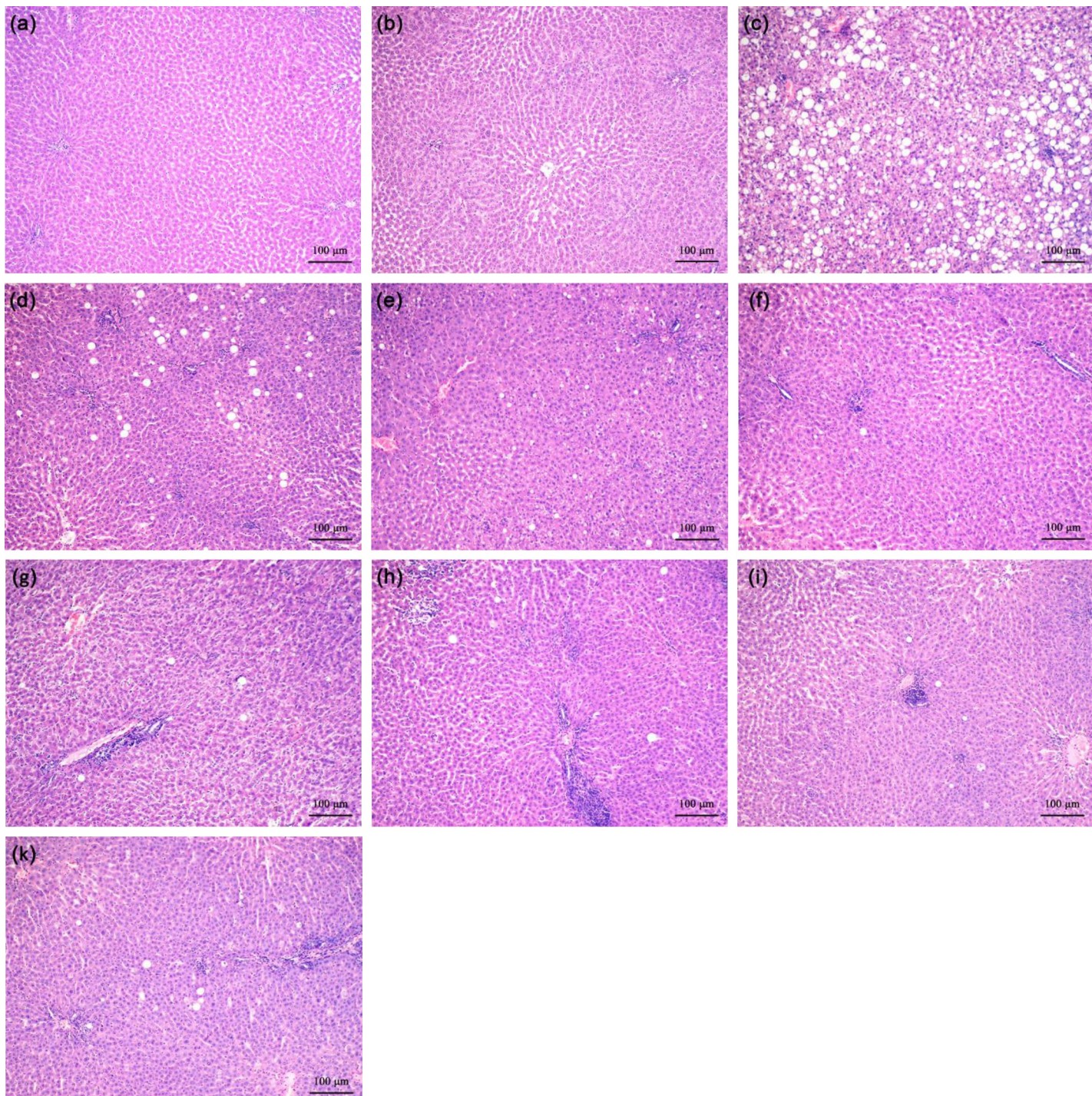

**Fig 6. Representative histological pictures of liver sections. (a)** Control. **(b)** Isocaloric + Solvent. **(c)** EtOH + Solvent. **(d)** I-ol (0.4 mg/kg b.w.) + EtOH. **(e)** I-ol (4 mg/kg b.w.) + EtOH. **(f)** I-ol (40 mg/kg b.w.) + EtOH. **(g)** I-an (0.4 mg/kg b.w.) + EtOH. **(h)** I-an (4 mg/kg b.w.) + EtOH. **(i)** I-an (40 mg/kg b.w.) + EtOH. **(k)** UDCA (40 mg/kg b.w.) + EtOH. Nine sections of each liver were prepared–with six or eight animals per group. All sections were stained with H. E. (original magnification: 100. Each scale bar indicates 100 μm).

In Order to gain a better insight into morphological changes of the liver architecture, more detailed histopathological examinations for I-ol in medium and high concentrations were additionally performed (S5 Fig). There was increased protection of the liver against lipid accumulation, inflammatory damages and early fibrosis, although rats were known to continue to receive an ethanol-containing diet.

Not all parameters selected at the beginning of the animal experiment confirmed the therapeutic effect of the experimental substances. Some of these parameters did not even show differences between the ethanol group and the control groups. More detailed information is summarized in S7 Table

## Discussion

Despite of its known health risk, alcohol abuse is consistently high worldwide [41, 42]. The toxicity of excessive alcohol consumption to the liver and the progression to liver cirrhosis and finally to hepatocellular carcinoma (HCC) is well documented [43]. Liver transplantation is the last possible treatment option and the only cure for end-stage ALD [44, 45] provided donor liver is available to patients with such a type of 'self-inflicted' disease [46], patients do not relapse [47] and suffer from de novo malignancies [44]

There are currently no FDA-approved drugs available for the treatment of ASH—not to mention another manifestation of ALD (alcoholic liver disease) [48]. Only two evidence-based pharmacotherapies are known: Corticosteroids [49] (i. e. prednisolone) as first-line therapy and pentoxifylline [50] as second-line therapy for acute alcoholic hepatitis. These recommendations apply only to short-term therapies and not for a period of use longer than 28 days [49, 51]. In general, standard therapy for ALD has not changed in the past 40 years [52].

There are a number of conventional and natural medicines that have been tested for a treatment of alcoholic liver disease. Most of them have been studied in cell and animal models, but few in clinical trials [53–55]. Interestingly, some of these products showed an influence on the mRNA and/or protein expression and enzymatic activity level of CYP2E1 (S3 File). None of the compounds listed has been developed as an enzyme inhibitor of CYP2E1 but some are in clinical use as therapeutics diseases other than ASH. A critical examination of the current data indicated a lack of any clearly proven efficacy of currently available pharmacotherapies. These compounds showed only limited efficacy in some clinical studies [52, 56]. A recent meta-analysis of the pharmacological treatment of alcoholic liver disease by the Cochrane Collaboration even concluded that there is considerable uncertainty about the efficacy of any pharmacological intervention due to very low quality evidence [57].

Due to the lack of a drug approved for the treatment of ASH, we selected the naturally occurring bile acid ursodeoxycholic acid (UDCA) as a 'positive' control standard molecule for comparison with our drug candidates. Although UDCA has several hepatoprotective activities, it is currently only approved for the resolution cholesterol gallstones, primary biliary cirrhosis (PBC) and primary biliary cholangitis (PSC). Nevertheless, UDCA is considered 'panacea' for pharmacological treatment of a variety of hepatobiliary disorders [58] and some preclinical studies have shown a positive effect on alcoholic fatty liver disease (AFLD) [59, 60].

In this study we present a new therapeutic approach for the medical treatment of ASH by competitive inhibition of the enzymatic activity of CYP2E1. I-ol and I-an represent the responsible lead compounds of the chemical group of ω-imidazolyl-alkyl derivatives. Their mechanism of action is described by reversible complexation of an imidazolyl nitrogen with trivalent heme iron ion (III), hydrophobic interaction of the alkyl chain with the substrate access channel and hydrophilic interaction of the hydroxyl group with backbone oxygen atoms of residues 238 (Asn) and 239 (Val) in case of I-ol. In this way, both compounds combine the inhibitory

property of imidazole and the stereochemical property of flexible mid-chain fatty acids to increase drug target interaction.

HPβCD acts as a cage-like carrier system and is able to capture I-ol very effectively making it to a promising galenic formulation for further studies. A very strong prevention of ROS production by CYP2E1 at its source of generation should be superior to all ROS scavenging compounds such as S-adenosyl-L-methionine (SAM), N-acetylcysteine or Vitamin E and must be considered as central point of therapeutic action. The postulated reduced formation of toxic acetaldehyde explains the strong effect on lipid metabolism, especially on hepatic lipogenesis and serum triglyceride concentration, which is additionally suppressed by activation of SREBP-1 c and inactivation of PPAR-α in addition. The elevated liver enzymes, which are biological markers of liver damage, have been almost completely reduced to basal levels.

The liver of the treated rats was histochemically comparable to that of the healthy rats, which did not represent any of the fatal damage observed in the ASH group. All parameters analyzed in vivo were altered by I-ol and/or I-an in a very or even extremely significant manner towards normality. This significance was generally lower for UDCA, which was unable to change the concentration of SRA, GSH, TBARS, LDL and VLDL. In addition, UDCA showed a weaker therapeutic effect ROS status, although its marked influence on this parameter and on liver histopathology was reported [61]. Regardless of positive results in some preclinical studies, clinical trials with UDCA for the treatment of alcoholic liver diseases showed unclear results [59, 62].

The network analysis mentioned confirmed that mortality at maximum follow-up was higher in the ursodeoxycholic acid group than in the no intervention group [57]. In addition, long-term use of high-dose UDCA (28–30 mg/kg/day) was associated with an increased risk of colorectal neoplasia in patients with ulcerative colitis and PSC [63]. The concentration in our studies was far above this value. Additional study results question the therapeutic value of UDCA for other biliary and liver diseases and therefore do not allow a clear statement as to whether UDCA has a therapeutic effect on such diseases [64, 65]. It is obvious that UDCA and the lead compounds identified in our study act by different mechanisms. In contrast to UDCA, which does not act in a single target-based manner [66] the lead compounds I-ol and I-an were designed to interrupt the pathobiochemical pathway mediated by CYP2E1 by direct ligand-target interaction. Therefore, it is expected that the therapy of a ROS-based disease such as ASH will benefit more from such a target-based approach.

Although I-ol reduced the enzymatic activity of CYP2E1 and the subsequent ROS generation more than I-an in in-vitro assays, both compounds showed comparable therapeutic effects in rats without statistically significant differences. These effects were observed throughout, although ethanol was administered continuously at high daily doses. Only in one case, hepatic cholestero, I-ol showed no therapeutic effect. A linear dose-response correlation of both compounds was observed for GSH, TBARS and GPx. I-an showed this additionally for SRA, hydrogen peroxide, concentration of serum lipids and γ-GT. I-ol exercised a linear dose-response correlation for GR and an inverse correlation in the reducing ALT activity. We have no explanation for this discrepancy, but it should be considered that the hydrogen group at the end of alkyl chain of I-ol may not only be responsible for the improved binding to CYP2E1, but also for a better pharmacological side effect profile. This fact will be relevant for further studies on the target specificity of the lead compounds, in particular with respect to the most important drug metabolizing CYP 450 proteins (i. a. 3A4, 2C9, 2C8 and 1A2) [67].

Although the animal model used led to an increase in the proportion of connective tissue in the liver, this was not significant. Therefore, it rather reflects an early stage of ASH. In addition, the CYP2E1 inhibitors were administered at the same time as the induction of liver damage by alcohol and not only after full development of the disease. These limitations should be

considered in more extensive studies to ensure a better transferability to the human clinical picture.

Proceeding towards a preclinical development requires a deeper understanding of long-term administration and differences in toxicological profiles between I-ol and I-an. Regardless of a successful development of the lead compounds into a new drug, this work defines CYP2E1 as a promising drug target for the therapy of ASH.

## Supporting information

**S1 Fig. Multiple alignment of CYP2E1 sequences.** These sequences originate from the species human, mouse, cow, pig, rabbit and rat. The sequence homology is compared with structural components of the active site A and the substrate access channel C. Colors encode the following: green (hydrophobic residues), red (acid), blue (lysine, arginine, histidine), orange (asparagine, glutamine), pink (serine, threonine).
(TIF)

**S2 Fig. CYP2E1 inhibition by data base inhibitors.** To ascertain whether an imidazole containing compound is already commercially available and better suited for development as a drug can-didate, we performed a large virtual library screening of the ZINK database. We found 14,738 hits. The 5,261 commercially available compounds could be classified as 4,710 synthetic and 551 natural molecules. After docking and final scoring process, the seven best synthetic and the three best natural candidates were purchased and tested in vitro for their potential to inhibit CYP2E1 activity. Different concentrations of these compounds dissolved in DMSO were added to 110 μM p-Nitrophenol in 100 mM HEPES buffer pH 7.6 and SUPER-SOMES™ with 50 nM human CYP 2E1 to calculate relative activity values. All ten compounds showed an impres-sively weaker inhibitory effect than I-ol, whereby the natural compound STOCK1N-69212 showed the strongest inhibitory effect among all other library compounds with an activity rate of 48.1% at a final concentration of 20 μM. The real inhibition potential of STOCK1N-69212 must be weaker because the solvent DMSO has an IC50 value of 0.065% (v/v) (doi: 10.6084/m9.figshare.12387107, https://figshare.com/s/f3ec912565aee777ea06), i. e. 20 μM corresponds to a DMSO volume of 0.02% (v/v).
(TIF)

**S3 Fig. Docking to CYP2E1 crystal structures as a backchecking analysis.** We calculated the median of predicted affinities of the best poses for each ex-perimental substance binding to all six protein structures as provided by default output. After calculating the rank for each conformation, the rank sum for each compound was calculated for the 'open protein conformations', the 'closed protein conformations' and both conformation types. Although there were no major conformational changes in the protein backbone, ligands occupied a versatile binding site, whose shape is determined by the rotameric states of Phe 298 and Phe 478. In crystal structures with imidazolyl-octanoic acid, -decanoic acid and -dodecanoic acid as stabilizing ligands, the alkyl chain occupied a hydrophobic channel of protein helices which was exposed by a rotation of the Phe 298 side chain. This led to a switch of the 'closed protein conformation' to the 'open protein conformation'. For the 'closed protein conformations', the alkyl derivatives showed a nearly constant binding affinity irrespective of their chain length, while the library compounds almost failed. On the other hand, there was a clear influence of the alkyl chain length on affinity to the 'open protein conformations', with the functional group -COOH increasing the affinity with 13cooh as the best scored compound. I-phosphocholine with its bulky head group was one of the worst binding compounds independent of protein conformation.
(TIF)

**S4 Fig. Schematic illustration of the interaction between I-ol und HPβCD. (a)** Wire Frame Mode of both molecules. **(b)** Stick Mode of I-ol and Surface Mode of HPβCD. **(c)+(d)** Surface Mode of both molecules; carbon of HPβCD (white), car-bon of I-ol (yellow), oxygen (red), nitrogen (blue). The illustrated conformation represents one of the most thermodynamically stable conformations of both in-teraction partners. The interior of the structure encloses a hydrophobic cavity of 260 nm diameter. The hydrophobic part of I-ol fits entirely into this cav-ity due to its flexible structure. The hydrophilic outer surface guarantees solubility in aque-ous environments (buffer systems, cell culture medium and blood). The hydroxyl group of I-ol generates polar interactions with hydroxyl groups on the outer sur-face of HPβCD (more pre-cisely: a hydrogen bond, which is not shown here). This bridge gives additional stability to the formed complex formed, which leads to a higher solubility of I-ol.
(TIF)

**S5 Fig. Additional histopathological pictures.** Ten sections were made from each liver—with six or eight animals per group. The magnification of the objective was 100-fold and that of the ocular 10-fold. The following are representative illustrations of these liver sections: **(a)** Portal tract of a healthy animal **(b)-(g)** Ethanol treated group **(b)** Necrosis near central vein (arrows point to apoptotic hepatocytes) **(c)** Massive layer of connective tissue (pink color) around the central vein lined with endothelium. **(d)** Microvesicular hepatocyte infiltration. Macrophages with large nucleoli (sign of intensive protein synthesis) that migrate into sinus. **(e)** Macrovesi-cular fatty dystrophy. The cell nucleus (arrow) is surrounded by massive lipid droplets **(f)** Enlargement of the sinusoidal lumen by destruction of the hepatocytes. Cluster of inflamma-tory leucocytes in the lumen. **(g)** Destruction of hepatocytes with cytotoxic lymphocytes. **(h)** Characteristic picture with significantly reduced destruction of liver tissue in the periportal area, microvesicular fatty infiltration and individual small lymphocyte aggregates in sinusoids by administration of I-ol (4 mg/kg b.w.). Comparable results were obtained with I-ol (40 mg/kg b.w) with only single small aggregates in sinusoids, neither apoptotic cells nor infiltrations of the portal and perivenular area with macarophages or lymphocytes. Each scale bar indicates 100 μm.
(TIF)

**S1 Table.** Scheme of animal studies (a) Group differences: Rats were divided into two control, one disease group and seven treatment groups. The disease group and the treatment groups received a diet with 5% (m/m) ethanol. (b) Time frame of the experiment: Ethanol feeding was introduced gradually over an adaption period of 2 weeks prior to the start of the experiment. The tested compounds and Hypromellose were administered daily from the beginning of etha-nol feeding until the end of the experiment.
(TIF)

**S2 Table. Composition of the liquid Lieber-De Carli (LDC) liquid diet (Lieber & De Carli, 1994).**
(TIF)

**S3 Table. Calculation of identity between overall sequences and the active site.**
(TIF)

**S4 Table. Enzyme kinetic data of ω-imidazolyl-alkyl derivatives.** Nonlinear fitting to calcu-late $K_i$ values was achieved using competitive inhibition as model including an offset in the case of I-phophocholine. $K_d$ values were calculated by nonlinear fitting of the experimental data to an equation including one specific binding site. The calculated apparent inhibition constant diminished with increasing ionic strength (i.e. HEPES to phosphate buffer) and

depends on the way I-ol is solubilized (i. e. DMSO or cyclodextrin) as well. The $K_d$ value of 578 nM obtained from a titration of rat liver microsomes with I-ol is close to the catalytic inhibition constant obtained with SUPERSOMES™.
(TIF)

**S5 Table. All in vitro tested imidazole ring system containing compounds.**
(TIF)

**S6 Table. Semiquantitative evaluation of the histological examinations.** [a]Steatosis and ballooning: 0, none; 1, $\leq$ 25%; 2, 26–50%; 3, $\geq$ 51–75% of liver parenchyma. [b]Inflammation: 0, none; 1, $<$ 5 signs of inflammation; 2, $>$ 5 signs of inflammation in the microscopic field at a 40x magnification.
(TIF)

**S7 Table. Not depicted and discussed parameters of the animal study.**
(TIF)

**S1 File. Synthetic library compounds.**
(XLSX)

**S2 File. Natural library compounds.**
(XLSX)

**S3 File. Therapeutic compounds related to CYP2E1.**
(XLSX)

## Acknowledgments

We are grateful to Prof. Dr. A. Cederbaum (Icahn School of Medicine at Mount Sinai, NY) for assistance with different HepG2 cells lines. We also thank Dr. Shiva Eisele-Kermani for assistance with linguistic revision, that greatly improved the manuscript.

## Author Contributions

**Conceptualization:** Torsten Diesinger, Vyacheslav Buko, Thomas Wirth, Thomas Haehner.

**Data curation:** Torsten Diesinger, Vyacheslav Buko, Alfred Lautwein, Radovan Dvorsky, Elena Belonovskaya, Oksana Lukivskaya, Elena Naruta, Siarhei Kirko, Thomas Wirth.

**Formal analysis:** Alfred Lautwein, Radovan Dvorsky, Elena Belonovskaya, Oksana Lukivskaya.

**Funding acquisition:** Torsten Diesinger, Thomas Wirth, Thomas Haehner.

**Investigation:** Alfred Lautwein, Radovan Dvorsky, Elena Belonovskaya, Oksana Lukivskaya, Elena Naruta, Siarhei Kirko, Viktor Andreev, Dominik Buckert, Sebastian Bergler, Christian Renz, Edith Schneider, Florian Kuchenbauer, Mukesh Kumar, Cagatay Günes, Berthold Büchele, Thomas Simmet, Dieter Müller-Enoch.

**Methodology:** Vyacheslav Buko, Elena Belonovskaya, Oksana Lukivskaya, Elena Naruta, Siarhei Kirko, Viktor Andreev, Dominik Buckert, Sebastian Bergler, Christian Renz, Edith Schneider, Florian Kuchenbauer, Mukesh Kumar, Cagatay Günes, Berthold Büchele, Thomas Simmet, Dieter Müller-Enoch.

**Project administration:** Torsten Diesinger, Vyacheslav Buko, Thomas Haehner.

**Software:** Radovan Dvorsky.

**Supervision:** Torsten Diesinger, Vyacheslav Buko, Thomas Wirth, Thomas Haehner.

**Validation:** Vyacheslav Buko, Alfred Lautwein.

**Visualization:** Alfred Lautwein.

**Writing – original draft:** Torsten Diesinger, Vyacheslav Buko, Alfred Lautwein, Radovan Dvorsky.

**Writing – review & editing:** Torsten Diesinger, Alfred Lautwein.

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
