## [Decision Letter · Decision Letter 0]

27 May 2020

PONE-D-20-13905

Drug targeting CYP2E1 for the treatment of ASH

PLOS ONE

Dear Dr. Diesinger,

Thank you for submitting your manuscript to PLOS ONE. After careful consideration, we feel that it has merit but does not fully meet PLOS ONE’s publication criteria as it currently stands. Therefore, we invite you to submit a revised version of the manuscript that addresses the points raised during the review process.

As you can see, both reviewers appreciated your work, but also listed several important points, that should be addressed during the revision.

We look forward to receiving your revised manuscript.

Kind regards,

Pavel Strnad

Academic Editor

PLOS ONE

Journal Requirements:

2. Thank you for your ethics statement: 'The animal study was conducted at the Institute of Biochemistry of Biologically Active Substances, Department of Biochemical Pharmacology. This is a member of the National Academy of Sciences of Belarus. The responsible ethics committee approved this study and follow-up studies. According to the disclosed protocol in the submitted manuscript, the rats were anesthetized by intraperitoneal injection of a five percent phenobarbital solution and then sacrificed by cutting the abdominal aorta.'

(a) Please amend your current ethics statement to include the full name of the ethics committee/institutional review board(s) that approved your specific study.

(b) Once you have amended this/these statement(s) in the Methods section of the manuscript, please add the same text to the “Ethics Statement” field of the submission form (via “Edit Submission”).

3. Please provide additional information about each of the cell lines used in this work, including any quality control testing procedures (authentication, characterisation, and mycoplasma testing). For more information, please see http://journals.plos.org/plosone/s/submission-guidelines#loc-cell-lines.

4. We noted that several of your references did not auto populate and instead the manuscript contains the following  "Error! Reference source not found", please replace this with the appropriate reference during your next revision."

6. At this time, we request that you  please report additional details in your Methods section regarding animal care, as per our editorial guidelines:

(1) Please state the source and number of mice used in the study  

(2) Please provide details of animal welfare (e.g., shelter, food, water, environmental enrichment)

(3) Please describe the post-operative care received by the animals, including the frequency of monitoring and the criteria used to assess animal health and well-being.

Thank you for your attention to these requests."

7. At this time, we ask that you please provide scale bars on the microscopy images presented in Figure 6 and refer to the scale bar in the corresponding Figure legend.

Reviewers' comments:

Reviewer's Responses to Questions

**Comments to the Author**

1. Is the manuscript technically sound, and do the data support the conclusions?

Reviewer #1: Yes

Reviewer #2: Yes

2. Has the statistical analysis been performed appropriately and rigorously? 

Reviewer #1: Yes

Reviewer #2: Yes

3. Have the authors made all data underlying the findings in their manuscript fully available?

Reviewer #1: Yes

Reviewer #2: Yes

4. Is the manuscript presented in an intelligible fashion and written in standard English?

Reviewer #1: Yes

Reviewer #2: Yes

5. Review Comments to the Author

Reviewer #1: Dear Dr. Diesinger and colleagues,

This abstract discusses the manuscript “Drug targeting CYP2E1 for the treatment of ASH” for publication in PLOS ONE. The authors describe the generation of small compounds and their possible use in the treatment of ASH, targeting the key cytochrome CYP2E1. Scope of the manuscript appears to the evaluation of the compounds in a cell culture and in vivo model. In general the study appears technically sound, structured, well written and of general interest to the scientific field. Nevertheless, a couple of inconsistencies have to be addressed and we would herewith encourage the authors to revise those points before publication:

Figure2: In case there is no reference standard available, it might allow easier comparison if the values are normalized to control HepG2 instead of plotting Fluorescence Intensity units, as those might differ between equipment/ lab. Please add corresponding data for I-an and UDCA in cell lines. Did you perform Pharmacokinetic studies in vivo? As reference for the reader, how does the drug concentration in cell culture correlate to serum/ in vivo concentrations?

Figure 5: ALT/ AST values appear very low compared to literature, please check if the shown measurements are correct.

Figure 6: Is the total magnification 100x? Please provide pictures with a comparable background (white) correction. Also for histological evidence of steatosis/ fibrosis we advise to perform an Oil red O/ Sirius red staining which would be quantifiable.

Supplementary Table 6: The control value provided for TNF appears beneath those stated in other literature. In addition, the standard curve for the ELISA used for TNF evaluation appears to have limit of 100pg/µl. Please show the standard curve for this experiment.

To add to the functional assays performed in this study it appears reasonable to provide some gene expression/ western blot data regarding the downstream effects of the drug on CYP2E1.

HepG2 cell line: Please provide further information regarding HepG2 E47 in a supplement, e.g. PCR/ WB showing the successful transfection.

As a general remark, we understand the experimental setup of administering the drug daily alongside the LDC diet. Nevertheless, treating an organism while inducing ASH is hardly translatable to the human situation. Did you do perform any experiment where disease progression was ongoing for some time before the treatment was started? Could you speculate on how this would influence the efficacy of the treatment e.g. stop disease progression/ reverse steatosis … ?

In case those minor points are addressed in a minor revision, we believe the manuscript provides a sound basis for the further evaluation of ASH therapeutics.

Reviewer #2: The present manuscript (PONE-D-20-13905) by Diesinger et al. entitled “Drug targeting CYP2E1 for the treatment of ASH” focused on developing new small molecule inhibitors of Cyp2e1 enzymatic activity that could be utilized for treatment of alcoholic steatohepatitis (ASH). The authors first designed novel CYP2E1 inhibitors using rational drug design. Then, they tested two promising compounds in vitro and in vivo using a rat, Lieber-DeCarli diet-based model of alcoholic liver disease (ALD). Concomitant administration of the tested drug along with alcohol diet improved liver injury and steatosis. Overall, major findings are supported by the data. Formally, the manuscript is little hard to read and follow. Improvements would be helpful especially in the results section which read way too much as a list of change percentages and does not educate the reader about the data or their meaning. Figures are not well labeled and thus not informative unless one reads the text, which makes it harder to read and understand. I have several comments that can further improve this manuscript and are listed below:

Major comments:

1. The title of the article as well as some parts of the paper that relate to ASH are imprecise. The authors present data to support the effect of their drugs in alcohol-induced liver injury and steatosis, but not steatohepatitis. None of the markers of liver inflammation were assessed. Most importantly, the model the author used is rather a model of early stages of ALD.

2. One of my major concerns is why the authors chose to test the potentials drug in a preventive treatment dosing as opposed to using the drugs for reversing an established ALD. This would be more important information that is relevant to clinical practice. In my opinion, this is a major limitation of the present study.

3. What were the bodyweight changes during the treatment study?

4. What was the final liver weight and liver-to-body weight ratio?

5. Was diet intake measured and caloric intake the same? These data should be included in the manuscript.

6. Figures are not easy to understand and follow. To make figures more intuitive, the authors can label graphs and images with more details (e.g., parameters assessed, use graph titles, treatment, legends, etc.)

Minor comments:

1. Any abbreviation, such as ASH, should be avoided in the title to improve understanding of the general audience. But as noted above, ASH should not even be mentioned as a major focus of the paper.

2. Replace “steatosis hepatitis” with appropriate terms – page 3, twice.

3. In the introduction, liver enzymes are not spelled out to introduce the abbreviation. Also, the abbreviations are not the most commonly used ones in the liver literature.

4. Actually, at multiple places of the manuscript several abbreviations are not introduced. Too many abbreviations make the manuscript less well readable, and especially if the abbreviations are quite specific to the field, e.g. chemistry vs. hepatology.

5. Many references throughout the manuscript were not inserted; see words ‘Error” in the text.

6. Listing % values in figure legend is not very informative and should be omitted.

6. PLOS authors have the option to publish the peer review history of their article (what does this mean?). If published, this will include your full peer review and any attached files.

Reviewer #1: No

Reviewer #2: No

---

## [Author Response · Author response to Decision Letter 0]

11 Jun 2020

Dear editor, dear reviewers.

I would like to take this opportunity to thank you for the constructive way in which the manuscript was revised. By discussing the content of your remarks, the manuscript has gained in format and increased its conclusiveness. We have addressed every single point of your expert opinion. We hope that the result of our revision meets your requirements. 

With kind regards

Dr. Torsten Diesinger

---

## [Decision Letter · Decision Letter 1]

26 Jun 2020

Drug targeting CYP2E1 for the treatment of early-stage alcoholic steatohepatitis

PONE-D-20-13905R1

Dear Dr. Diesinger,

We’re pleased to inform you that your manuscript has been judged scientifically suitable for publication and will be formally accepted for publication once it meets all outstanding technical requirements.

Kind regards,

Pavel Strnad

Academic Editor

PLOS ONE

Additional Editor Comments (optional):

Reviewers' comments:

Reviewer's Responses to Questions

**Comments to the Author**

1. If the authors have adequately addressed your comments raised in a previous round of review and you feel that this manuscript is now acceptable for publication, you may indicate that here to bypass the “Comments to the Author” section, enter your conflict of interest statement in the “Confidential to Editor” section, and submit your "Accept" recommendation.

Reviewer #1: All comments have been addressed

Reviewer #2: All comments have been addressed

2. Is the manuscript technically sound, and do the data support the conclusions?

Reviewer #1: Yes

Reviewer #2: Yes

3. Has the statistical analysis been performed appropriately and rigorously? 

Reviewer #1: Yes

Reviewer #2: Yes

4. Have the authors made all data underlying the findings in their manuscript fully available?

Reviewer #1: Yes

Reviewer #2: Yes

5. Is the manuscript presented in an intelligible fashion and written in standard English?

Reviewer #1: Yes

Reviewer #2: Yes

6. Review Comments to the Author

Reviewer #1: We feel the Quality of the manuscript has greatly improved with the added Information in the figures. We would have liked to see further Experiments on the subject, but still believe the data included is sufficient for publication.

Reviewer #2: The authors have adequately addressed my comments and concerns, and made appropriate changes to the manuscript.

7. PLOS authors have the option to publish the peer review history of their article (what does this mean?). If published, this will include your full peer review and any attached files.

Reviewer #1: No

Reviewer #2: No

---

## [Editor Report · Acceptance letter]

8 Jul 2020

PONE-D-20-13905R1 

Drug targeting CYP2E1 for the treatment of early-stage alcoholic steatohepatitis 

Dear Dr. Diesinger:

I'm pleased to inform you that your manuscript has been deemed suitable for publication in PLOS ONE. Congratulations! Your manuscript is now with our production department. 

Kind regards, 

on behalf of

Dr. Pavel Strnad 

Academic Editor

PLOS ONE